# Computer-Based Drug Design of Positive Modulators of Store-Operated Calcium Channels to Prevent Synaptic Dysfunction in Alzheimer’s Disease

**DOI:** 10.3390/ijms222413618

**Published:** 2021-12-19

**Authors:** Lernik Hunanyan, Viktor Ghamaryan, Ani Makichyan, Elena Popugaeva

**Affiliations:** 1Laboratory of Molecular Neurodegeneration, Peter the Great St. Petersburg Polytechnic University, 195251 St. Petersburg, Russia; lernik.hunanyan@rau.am; 2Laboratory of Structural Bioinformatics, Institute of Biomedicine and Pharmacy, Russian-Armenian University, Yerevan 0051, Armenia; viktor.ghamaryan@rau.am (V.G.); ani.makichyan@rau.am (A.M.)

**Keywords:** nSOCE, TRPC6, in silico drug design, Alzheimer’s disease

## Abstract

Store-operated calcium entry (SOCE) constitutes a fine-tuning mechanism responsible for the replenishment of intracellular stores. Hippocampal SOCE is regulated by store-operated channels (SOC) organized in tripartite complex TRPC6/ORAI2/STIM2. It is suggested that in neurons, SOCE maintains intracellular homeostatic Ca2+ concentration at resting conditions and is needed to support the structure of dendritic spines. Recent evidence suggests that positive modulators of SOC are prospective drug candidates to treat Alzheimer’s disease (AD) at early stages. Although STIM2 and ORAI2 are definitely involved in the regulation of nSOC amplitude and a play major role in AD pathogenesis, growing evidence suggest that it is not easy to target these proteins pharmacologically. Existing positive modulators of TRPC6 are unsuitable for drug development due to either bad pharmacokinetics or side effects. Thus, we concentrate the review on perspectives to develop specific nSOC modulators based on available 3D structures of TRPC6, ORAI2, and STIM2. We shortly describe the structural features of existing models and the methods used to prepare them. We provide commonly used steps applied for drug design based on 3D structures of target proteins that might be used to develop novel AD preventing therapy.

## 1. Introduction

Alzheimer’s disease (AD) is considered the main cause of elderly dementia. Medications currently used in clinics (blocker of acetylcholinesterase and NMDAR antagonists) demonstrate only temporal relief of the symptoms [1]. The population continues to age; thus, the prevalence of AD increases. There is an urgent need in the search for an effective disease-preventing drug.

Search of the disease-modifying therapy should be based on the normalization of intracellular mechanism that leads to the manifestation of the disease’s hallmark. A pathological hallmark of the AD is progressive memory loss that is well correlated with synaptic loss and dysfunction [2,3,4].

We have recently shown that synaptic loss in AD might be caused by dysfunction in fine-tuning mechanism—store-operated calcium entry in hippocampal neurons [5,6,7,8,9,10]. Neuronal store-operated calcium entry (nSOCE) is downregulated in response to ER Ca^2+^ overload. nSOCE is needed to support the activity of pCaMKII and is suggested to be necessary to protect synapses from amyloid and presenilin-mediated toxic effects especially at rest [8,11,12]. It has been demonstrated that store-operated calcium entry in hippocampal neurons is regulated by tripartite complex: STIM2—protein that resides in ER and senses the drop in ER Ca^2+^ concentration and TRPC6/ORAI2—plasma membrane channel complex that in response to STIM2 activation delivers Ca^2+^ ions via the cellular membrane into the cytoplasm [8]. Moreover, multiple evidence suggest that the genetic modification or pharmacological corrections of the nSOCE pathway demonstrate a therapeutic effect in preclinical models of AD [5,6,7,8,9,10,13,14]. However, the compounds that are positive modulators of nSOCE that have been studied so far are unsuitable for preclinical studies due to complications in their pharmacokinetic profile and bioavailability (reviewed recently in [15,16,17]).

STIM2, ORAI2, and TRPC6 proteins are potential molecular targets for the development of AD-preventing therapy. Within the current review, we will discuss the prospectives to develop the specific nSOCE targeting pharmacological agents. It is important to note that nSOCE might be either downregulated or upregulated depending on the pathogenetic model used to study AD. However, there is no preclinical model available so for that would allow testing the cognitive benefit of usage of nSOCE antagonists. Thus, within the current paper, we will focus on the development of positive modulators of the nSOCE intracellular signaling pathway.

Historically, the drug discovery process was a trial and-error testing of chemical substances on animals and matching the apparent effects to treatments [18]. Despite the direct assessment of a compound-mediated cognitive benefit in animal models of AD, this approach has a major disadvantage in that it does not provide knowledge on the molecular target. Thus, when switching to clinical investigation, compounds developed via a historically standard pharmacological approach usually show either many cross-specificities and consequently toxicities or the absence of therapeutic effects. Moreover, this approach is time and cost consuming.

In contrast, in silico drug design begins with a knowledge of specific chemical responses in the body and tailoring combinations of these to fit a treatment profile [18]. The major advantage of in silico drug design is the search and design of chemicals that will normalize either the structure or function of the target protein. In addition, in silico approach saves the time and money of government or pharma investors.

We concentrate the review on perspectives to develop specific nSOC modulators based on the available 3D structures of TRPC6, ORAI2, and STIM2. We shortly describe the structural features of the existing models and methods used to prepare them. We provide commonly used steps applied for drug design based on 3D structures of target proteins that might be used to develop novel AD, preventing therapy.

## 2. Principles of Computer-Based Drug Design

Improvements in experimental methods for the structural identification of biomacromolecules and chemical compounds, particularly crystallographic and X-ray structural methods, made it possible to work with high-resolution molecular models [19]. On the other side, bioinformatics and big data analysis are rapidly developing [20], suggesting to the scientific community new methods and tools to process biological information [21].

Nowadays, the development of a drug compound by using computer technology is achieved by the use of a combined approach involving a number of multistage processes [22]. These processes include the following: virtual screening, which is a compound selection process that calculates structure similarity values based on quantitative structure–property (QSPR) and structure–activity (QSAR) characteristics, using different similarity descriptors [23]; pharmacophore design, which is a procedure for determining the sets of a compound’s spatial–energy characteristics necessary to ensure high-affinity complexation of the compound with the biological target, leading to a change in the target’s biological response [18,24]; dynamic modeling is one of the main techniques that allows one to simulate the interactions of the target–ligand system in real time [25]; chemogenomics and proteomics are methods used for the design of pharmaceutical compounds, and researchers are studying genomic and/or proteomic responses to compounds of various natures in biomedical molecular systems [26]. Along with these approaches, quantum–chemical and biophysical methods of control and the verification of supramolecular processes obtained in silico are also being intensively used [27].

## 3. Neurophysiology of STIM2 Protein

STIM2 was initially described as a protein that regulates store-operated calcium entry in non-excitable cells; later on, it has been shown that it plays a major role in spine stabilization and regulates nSOCE in neurons [6].

STIM2 is a predominant isoform in the hippocampus [6,8]. In a model of focal cerebral ischemia, the genetic deletion of STIM2 protein disrupted nSOC and conferred protection from neurological damage [28]. Korkotian et al. suggested that STIM2 moves to dendritic spines and regulates Orai1-mediated nSOC [29]. STIM2, but not STIM1, influences the formation of excitatory dendritic spines and shapes basal synaptic transmission in excitatory neurons [30,31]. There is an observation that STIM2 functions in an nSOC-independent manner by regulating the phosphorylation and surface expression of the AMPAR [30]. It has been reported that STIM2 is required for the stable expression of both long-term potentiation (LTP) and long-term depression (LTD) at CA3–CA1 hippocampal synapses [31]. Our laboratory reported that virus-mediated knockdown of STIM2 protein expression causes mushroom spine shrinkage and causes a loss of nSOC in hippocampal dendritic spines [6]. Moreover, it has been observed that STIM2 hyperexpression as well as the pharmacological activation of nSOC in the hippocampus is able to protect mushroom spines in different models of Alzheimer disease pathology [5,6,8]. In addition, the downregulation of STIM2 proteins was observed in cells from Alzheimer’s disease (AD) patients and in AD mouse models [6,32]. Recently, STIM2 has been identified to bind EB3 in hippocampal neurons. It has been shown that STIM2–EB3 interactions participate in microtubule movements and play a role in the stabilization of mushroom spines [33]. There are experimental evidence that STIM2 has an nSOC-dependent and nSOC-independent function in the brain. In the following section, we will try to describe the intracellular signaling pathways that contribute to different STIM2-dependent neurophysiology and discuss the possibility of developing STIM2-specific pharmacological modulators.

## 4. STIM2 as Pharmacological Target

Multiple evidence suggests that STIM2 contributes to a broad array of fundamental physiological processes (recently reviewed in [34]), making it a difficult target for a drug design. For example, the upregulation of STIM2 protein expression has been observed in cancer cells [35]. The increased activity of STIM/ORAI complexes is possible because of vascular remodeling and consequently heart failure. Thus, talking in terms of pharmacological modulators of STIM2 activity, the majority of the literature is devoted to the search and development of STIM2 antagonists [34,36,37,38].

There are only a couple of papers describing possible positive unspecific modulators of STIM2 activity. In vascular smooth muscle cells, it has been observed that L-type Ca^2+^ channel blockers—dihydropyridines, phenylakylamines, and benzothiazepines activate STIM proteins by acting on a 10-amino acid N-terminal region located in the endoplasmic reticulum lumen [39], promoting the formation of STIM1 and 2/ORAI puncta that leads to the upregulation of Ca^2+^ influx. Excessive Ca^2+^ influx via STIM/ORAI causes heart failure due to vascular remodeling [39]. Whether L-type Ca^2+^ channel blockers have a similar cross-specificity to STIM proteins in the brain is not clear. Another study shows that a low concentration of 2-APB (chemical with a broad spectrum of cross specificity) potentiates SOCE exclusively through STIM2 in non-excitable cells [40].

The pharmacology of STIM2 is further complicated by the existence of three different post-translational modifications of STIM2 at least in HEK293T cells [41]. Isoforms of STIM2 differ in the length of the signal peptide at the N-terminus, are localized at different cellular compartments, and are responsible for the modulation of different intracellular signaling pathways [41]. The dominant isoform of STIM2 without a signal peptide resides in ER and participates in SOCE regulation; preSTIM2 (with full signal peptide) is localized near the plasma membrane and regulates store-independent functions of ORAI1 channel, the third isoform with a truncated signal peptide plays a role in transcription regulation [41]. Later on, splice variant STIM2β has been observed to be expressed in different human tissues including the brain [42]. It has been shown that STIM2β downregulates ORAI1-mediated SOCE in HEK293 and immune cells [42,43].

Despite the beneficial effects of STIM2 protein expression for the stabilization of mushroom spines and nSOCE support in hippocampal neurons, STIM2 should not be taken as a lead cellular target for drug design due to the presence of preSTIM2 isoforms that regulate the activity of the ORAI channel in the store-independent manner as well as STIM2β, which downregulates ORAI1-mediated SOCE.

## 5. Structural Model of STIM2

There is only one NMR structure of calcium-loaded EF-SAM STIM2 (identification number 2L5Y) [44]. The model is presented as a chain A monomer with a sequence length of 143 amino acid residues with a molecular weight of 17.87 kDa. The presented protein data bank (PDB) file includes the values of the spatial parameters of atoms for amino acid residues in positions from 62 to 205. The structure was used to identify the difference in oligomerization state between STIM2 and STIM1 [44]. There is no ligand (antagonist or agonist)-bound structure of STIM2 available, making it impossible to perform structure-based drug design.

## 6. Neurophysiology and Pharmacology of ORAI2 Channels

ORAI2 as well as its homologues ORAI1 and 3 were discovered by genome-wide RNAi screens for SOCE inhibition in patients with immune deficiency and CRAC channel dysfunction [45,46,47]. ORAI2 is not well-defined functionally due to the lack of patients with null mutations, gene-deficient mouse models, and selective inhibitors of individual ORAI homologues [48]. It is important to note that ORAI2 is prominently expressed in murine but not in human brain tissue [49,50,51,52].

So far, only one chemical compound, Synta66 (N-[4-(2,5-dimethoxyphenyl)phenyl]-3-fluoropyridine-4-carboxamide), has been shown to potentiate ORAI2-dependent calcium release-activated calcium (CRAC) channel current density in HEK293T cells overexpressing CFP-tagged ORAI2 on the ORAI1/2/3 null background [53].

Nevertheless, the shRNA-mediated knockdown of ORAI2 in primary hippocampal cultures causes a significant reduction in nSOCE [8]. Surprisingly, the overexpression of ORAI2 did not increase but attenuated hippocampal nSOCE [8]. A similar inhibition of SOCE by ORAI2 overexpression was also observed in HEK293T cells [54]. Orai2 has been shown to function as a negative modulator of SOCE in various other cell lines such as chondrocytes [55], Jurkat T cells [56], mouse T cells [57], primary amenoblasts [58], and human neuroglioma-derived cells [59]. In naïve T cells, the genetic deletion of ORAI1 reduces SOCE, whereas the deletion of ORAI2 enhances. It has been suggested that in immune cells, ORAI2 forms heteromeric channels with ORAI1 and fine-tunes the magnitude of SOCE [57]. In hippocampal neurons, the fine-tuning properties of ORAI2 were not reported. However, it has been suggested that ORAI2 is able to form functional nSOCE only in the presence of STIM2 or/and TRPC6 [8].

The development of specific ORAI2 modulators is further complicated by the absence of appropriate structural information including ORAI2 homologous models available on the SwissProt and TrEMBL databases.

## 7. TRPC6 as a Pharmacological Target

The literature data indicate that TRPC6 channels may represent an attractive molecular target for the development of therapy that slows down AD. There is also genetic evidence that TRPC6 is involved in AD pathogenesis. The decreased expression of TRPC6 mRNA was observed in blood [60], in leukocytes [14] from patients with AD, and moderate cognitive impairment as well as in AD patient-specific iPSCs [13]. Knockdown of TRPC6 expression blocks nSOCE in hippocampal neurons. The overexpression of TRPC6 channels or their pharmacological activation restores nSOCE and the loss of spines in hippocampal neurons in AD [8,9]. Mice that overexpress TRPC6 in the brain show improved cognitive function and increased excitatory synapse formation [61].

TRPC6 can be activated in a receptor-operated mode through the stimulation of G-protein coupled receptors and the synthesis of a secondary messenger diacylglycerol (DAG) that binds to TRPC6 and helps to open the channel for the entry of calcium ions [62]. TRPC6 can be also activated via store-operated mode when inositol 1,4,5-trisphosphate (IP3) or some other signal releases Ca^2+^ from the ER stores [63].

## 8. Possible Side Effects of Positive Modulation of TRPC6 Channel Activity

The expression of TRPC6 is observed in various tissues and organs of the human body. Mutations in TRPC6 leading to an increase in its function as a channel for conducting calcium ions have been implicated in the pathogenesis of focal segmental glomerulosclerosis [64]. In addition, TRPC6-dependent signaling pathways are involved in the development of various types of cancers, as well as in the dysfunction of cells of the immune system [65]. The hyperactivation of TRPC6 channels by hyperforin leads to gastrointestinal disorders [66]. Long-term exposure to positive TRPC6 channel modulators may have side effects associated with impaired renal function (proteinuria), with decreased immune response. There is a possibility that there are contraindications to the use of positive TRPC6 modulators in patients at risk of the onset and progression of cancer.

## 9. Piperazines as Modulators of TRPC6 Channel Activity

Various chemical structures that are capable to activate TRPC6 channels are described in the literature. Among them are hyperforin [67], NSN21778 [8], and piperazine derivatives [68]. Within the current review, we will not describe all positive modulators of TRPC6 channels known today, since they have been recently described by our research group [15]. We will concentrate on the description of disubstituted piperazines as potential positive modulators of TRPC6 channels. We are particularly interested in piperazine derivatives, since they are small molecules that are widely used in modern medicine to treat neurological disorders such as depression, schizophrenia, and epilepsy [69], indicating that these drugs are able to penetrate the brain–blood barrier (BBB) and are well tolerated by patients.

Piperazine derivatives as TRPC6 modulators were first described by Sawamura et al. in 2016. The authors have shown that [4-(5-chloro-2-methylphenyl)piperazin-1-yl] (3-fluorophenyl)methanone (PPZ1) and 2-[4-(2,3-dimethylphenyl)-piperazin-1-yl]-N-(2-ethoxyphenyl)acetamide (PPZ2) activate TRPC6 channels in a DAG-dependent way, and the neuroprotective effect of these compounds is carried out by activating the brain-derived neurotrophic factor (BDNF) signaling pathway [68]. In addition, PPZ1 and PPZ2 are cross-specific for TRPC3 and TRPC7, complicating their use as a drug.

Later on, we have found an analog of PPZ2 in the InterBioScreen library, N-(2-chlorophenyl)-2-(4-phenylpiperazin-1-yl)acetamide (51164). We have shown that 51164 activates TRPC6 in store-operated mode, recovers mushroom spines, and induces LTP in brain slices from 5×FAD mice (B6SJL-Tg(APPSwFlLon, PSEN1*M146L*L286V)6799Vas/Mmjax, MMRRC Stock No: 34840-JAX) [9]. However, we observed that 51164 is unstable in plasma and does not penetrate the BBB (unpublished data).

Interestingly, disubstituted piperazine derivatives demonstrate antagonistic properties as well. Cycloalkyl-piperazinylethanol derivatives have been shown to inhibit SOCE (structure #39 [37]). It was reported that this compound was effective at low-micromolar concentrations and demonstrated high selectivity for the inhibition of store-operated compared to receptor-operated calcium channels [37]. We have observed that trifluoperazine, a disubstituted piperazine derivative drug used in clinic to treat schizophrenia [70], inhibits TRPC6 in HEK293T cells (unpublished data). Therefore, the size of the radicals that are located at the 1st and 4th position of the piperazine ring seems to be a limiting factor in determining the nature of piperazine derivatives-mediated modulation of TRPC6. In future drug design based on piperazine derivatives as positive modulators, it is important to identify the pharmacophore of the lead compound in order to develop target-specific drugs. In the following section, we will describe the published 3D structures of TRPC6 and provide commonly used computer-based steps applied for in silico drug design.

## 10. Structural Models of TRPC6

Based on their sequence features, TRPC6 channels have a tetrameric transmembrane pore formed by six transmembrane helices, similar to other TRP channels. In addition, they have a large cytoplasmic N-terminus that contains four ankyrin repeats and a C-terminal coiled-coil motif [71]. It is important to note that TRPC6 can form either homotetramers or heterotetramers with other members of TRP channels family with variable calcium ion permeability [72].

The www.RCSB.org (accessed date 1 October 2021) database contains five molecular models of TRPC6. All models were obtained using electron microscopy with ≤3.8 A resolution. From a structural point of view, the presented models are homotetramers. PDB ID: 6CV9 belongs to the Mus musculus organism and is represented as a cytoplasmic domain of a protein. The protein structure is presented in its native form without mutations. The 6CV9 molecular model was released in 2018. All other models shown in the PDB are human ones. It should be noted that TRPC6 is a potential drug target included in the FDA list, information about which is presented both on the Human Protein Atlas online resource (https://www.proteinatlas.org/ENSG00000137672-TRPC6 (accessed date 1 October 2021); [73]) and in the Therapeutic Target Database under identification number T80165 [74].

The first complete molecular model of TRPC6 (PDB ID: 5YX9, Table 1) was uploaded to PDB in May 2018. The model consists of 931 amino acid residues. The molecular weight of the homotetramer is 425.81 kDa. This model is affixed without mutations. The dimensions of the described tetramer in three-dimensional space are 75 × 75 × 150 Å. From the architectural point of view, the model consists of an intracellular cytoplasmic domain (ICD) and a transmembrane domain (TMD) [71]. The cytosolic N-terminus consists of four repeats and nine linker helices (Linker Helix). The transmembrane domain is represented by six helices. A cavity (pore) is formed between the 5th and 6th domains, consisting of two sections: P1 and P2. The transient receptor potential (TRP) helix is located immediately after the 6th domain and is located at the C-terminus of the cytosol along with the C-terminal helices 1 (C helix1) and 2 (C helix2). The last helices play an important role in interacting with other proteins [75]. Molecular models with ID numbers 6UZA and 6UZ8 were posted in 2020 describing the interaction of an N-terminally truncated (Δ2–72) human TRPC6 with either an antagonist (AM-1473) or an agonist (AM-0883) (Table 1) [76]. The length of the presented models is 847 amino acid residues. 6UZ8 is a model with a high resolution with a value of 2.84 Å compared to other models.

The 6UZ8 model is the first model describing the binding of an agonist with a tetrameric structure of TRPC6 [76]. Agonist AM-0883, which consists of a chloro-indole, a piperidine, and a benzodioxin, binds TRPC6 at TMD and occupies a groove between S6 (S6 from the 6UZ8 structure is homologous to the 6th domain from the 5YX9 structure based on T-coffee alignment results http://tcoffee.crg.cat/apps/tcoffee/result?rid=7807049f accessed date 8 October 2021) of one subunit and the pore helix of the adjacent subunit. Bai et al. has shown that the agonist forms hydrophobic interactions with Phe675 and TRP680 on the pore helix and Tyr705, Val706 and Val710 on S6. The authors suggest possible hydrophilic interactions between the indole ring of AM-0883 and Glu672 and Asn702 [76]. Previously, the conserved LFW motif (residues 678–680 in TRPC6) on the pore helix has been identified to be essential for channel activation [72,77]. Substitution of all three residues with alanine in TRPC6 resulted in nonfunctional channels without altering plasma membrane expression [77].

## 11. Molecular Docking

Currently, molecular modeling methods and molecular docking in particular are intensively used in modern pharmaceutical chemistry for research and primary assessment of the bioactivity of compounds [25]. The use of this method leads to an understanding of the types and mechanisms of action of the compounds with a possible interaction with the target [79]. Molecular docking is used for the conformational search for the best and most reliable ligand (compound) orientation during ligand–target complexation. To achieve the maximum result, paired docking is usually used—the prediction of the interaction of the ligand with the target. Nowadays, several types of molecular docking are used. One of them is “flexible” docking [80], where the maximum possible number of degrees of freedom is determined for the ligand, while individual elements of the target have an ultimate number of degrees of freedom. The second type is “hard” docking [81], in which the maximum possible number of degrees of freedom is determined for the ligand, and the target is fixed. If in molecular docking, there is an understanding about the structural and functional features of the target, then the methodology of “site-directed” docking is applied [82]. In the absence of information regarding the possible binding sites of the target (active site, certain binding sites, etc.), the methodology of “blind” docking is applied [83]. It should be noted that the achievement of statistical reliability when using blind docking is carried out by repeating the experiment ≥100 times with the analysis of the spatial and energy properties of the conformers. If a flexible compound with several rotatable bonds is presented as a ligand, then the repeatability of the experiment increases by several times [84]. One of the most important criteria for molecular docking is the choice of an algorithm for searching the ligand’s best conformation on the target surface. Nowadays, several search algorithms are used, one of which is the Lamarckian genetic algorithm (LGA) [85], by which each generation of conformers is accompanied by energic minimization at a definite spatial point. The AUTODOCK 4.2.6 program is an example [86], which is based on LGA. The Tabu search algorithm (or meta-algorithm of local search) [87] makes it possible not to stop at a local optimum point, allowing you to move from one local optimum to another to find the best global optimum. The PAS-Dock v. 1.0 and PSI-Dock v. 1.0 [88] programs are the ones that use this algorithm. One of the common search algorithms used in molecular docking is the Monte Carlo algorithm [89]. It uses the “annealing” method [90], in which the criteria and conformer values are calculated after each iteration. As a result, using the Monte Carlo method, the best spatial orientation is determined at a high value of the free energy. Representatives on this basis of this algorithm are the programs MCDOCK v. 1.0 [91] and ICM v.3.7-2 [81]. There is also a Fast Shape Matching Algorithm called “Fragmentary Docking” [92]. The FlexX v.1.8 software package uses this algorithm [93].

### 11.1. Docking and Conformation Analysis of 51164 Compound (Own Study)

As an example of how docking and conformational analyses (see Materials and Methods section) are performed, we describe our own study of the 51164 compound. Despite its bad pharmacological profile, we decided to perform docking and conformational analyses of the 51164 compound in order to identify interaction cites with TRPC6. As a positive control, we took hyperforin, which is a widely used positive modulator of TRPC6 [67]. The spatial–energy parameters of the interaction were calculated, and a conformational map of complexation was constructed, which made it possible to determine the type and nature of the interaction, as well as to identify the amino acid residues involved in the complex formation.

The obtained spatial and energy parameters of the interaction indicate that hyperforin and 51164 interact with the Ca^2+^ permeable pore helix ([71,76]) of the TRPC6 protein with binding energies of −7.7 ± 0.38 and −7.1 ± 0.35 kcal/mol, respectively. The binding constants for both interactions were calculated, which are equal to 4.6 × 10^5^ for hyperforin and 1.7 × 10^5^ for 51164. The obtained results of conformational analysis indicate that hyperforin and 51164 interact with ALA404, LEU411, PHE443, ILE610, ILE613, LEU614, and ASN617 amino acids with different type of interactions (summarized in Table 2, Figure 1). We identified that hyperforin and 51164 bind to the similar region of the TRPC6 pore that has been previously reported for TRPC6 channel agonist, AM-0883 [76]. The prevalent type of interactions for both structures with TRPC6 is hydrophobic (a similar dominant type of interaction is observed for the AM-0883 structure [76]) (Table 2). A Van der Waals type of interaction is absent, but a donor acceptor type of interaction is present (two hydrogen bonds) in a hyperforin-bound TRPC6 structure (Figure 1, Table 2). Pi-Sigma, a hydrophobic type of interaction is present in a 51164 bound TRPC6 structure (Figure 1, Table 2). A different type of interaction of hyperforin and 51164 with amino acids forming a calcium-permeable pore of TRPC6 may explain the different mode of TRPC6 activation mediated by hyperforin (direct activator) [67] and 51164 (DAG and store-dependent activator) [9]. Docking and conformation analysis of the 51164 compound will be taken as key characteristics of the pharmacophore.

### 11.2. Future Steps in Drug Design Based on 51164 Structure

One of the most important stages of rational drug design is to increase the bioavailability characteristics of compounds based on the calculation of absorption, distribution, metabolism, excretion, and toxicity parameters that can be done by the freely available online software packages SwissADME (http://swissadme.ch/ accessed date 8 November 2021) and admetSAR (http://lmmd.ecust.edu.cn/admetsar2/ accessed date 8 November 2021). We will use Swiss-Similarity platforms (http://swisssimilarity.ch/ accessed date 8 November 2021); SIMCOMP (https://www.genome.jp/tools/simcomp/ accessed date 8 November 2021) for virtual screening based on the structure of disubstituted piperazines.

It is important to determine the pharmacophore (in our case, key characteristics of the pharmacophore are listed in Table 2) on the basis of the structure of the target’s active center and conduct a search procedure in freely available chemical databases.

Using the resources of SwissBioisostere (http://www.swissbioisostere.ch/ accessed date 8 November 2021), FragVLib (https://www.bioinformatics.org/fragvlib/ accessed date 8 November 2021) will help to carry out a directed fragmentary design and determine compounds close in spatial and energy characteristics to the pharmacophore. The determination of the toxicity of compounds is known to play a key role in drug design. There are various online software packages and modules that predict toxicity in silico, such as the TOXtree (http://toxtree.sourceforge.net/predict/ accessed date 8 November 2021), Gusar tox (http://www.way2drug.com/gusar/acutoxpredict.html accessed date 8 November 2021) MetaPred (http://crdd.osdd.net/raghava/metapred/ accessed on 8 November 2021), and LAZARtox (https://lazar.in-silico.ch/predict accessed date 8 November 2021) programs.

There are several databases such as https://www.ebi.ac.uk/chembl/, https://go.drugbank.com/, https://zinc.docking.org/ (accessed date for all databases 12 December 2021), which today are freely available and enable the user to work with a large chemical space. As a result of in silico studies, we will select the top compounds that meet all leadlikeness criteria, exhibiting high affinity for the active center of TRPC6. On the basis of in silico results, in vitro studies of the neuroprotective properties of the selected compounds, analysis of pharmacokinetic characteristics, and preclinical studies will be carried out.

### 11.3. Materials and Methods

The three-dimensional structure TRPC6 was taken from https://www.uniprot.org (accessed date 1 October 2021) with KB number: Q9Y210. Docking analysis was performed using Autodock vina v 1.1.2 software package [94]. Considering that the molecular model of TRPC6 is a homotetramer, Chain A of TRPC6 was chosen as a work model with subsequent optimizations. USFC Chimera [95] was used for stabilization and energy minimization. The molecular model of hyperforin was taken from https://pubchem.ncbi.nlm.nih.gov (accessed date 1 October 2021) with ID number CID: 441298. The model of 51164 was designed and optimized using MM2 force fields by us, which is the accepted practice [96]. The Chem Office v. 13.057 was taken for structural and energy optimization of 51164 [97]. The clustering of docking results and determination of the best conformers was carried out based on the Forel algorithm with Python. Visualization and conformational analysis of complexation was carried out using the BIOVIA Discovery Studio software v.20.1.0.19295. The binding constant was calculated based on the Poisson–Boltzmann equation. The docking box size did not exceed 27,000 Å^3^, and “exhaustiveness” was set to 1024, and the numbers of modes are 20 with 20 iterations for best mode calculation.

## 12. Conclusions

Experimental evidence (decreased TRPC6 mRNA expression in AD patient samples, improved cognitive function, and increased excitatory synapse formation in mice that overexpress TRPC6 as well as synaptoprotective properties of positive TRPC6 modulators) suggest that TRPC6 is the preferred molecular target in comparison to STIM2 and ORAI2. STIM2 and ORAI2 are not easy to target pharmacologically. Modulation of their activities most likely would bring more severe side effects, because STIM2 is involved in the regulation of many key physiological processes, while ORAI2 is functionally uncharacterized, and it needs further investigations and structure identification. The therapeutic effect of positive modulators of TRPC6-dependent nSOCE is the preservation of synapses structures that is believed to support memory storage in the aging AD brain. TRPC6 binding structures available today need further characterizations and optimization in order to improve their pharmacokinetic and bioavailability profiles. Computer-based drug design is a perspective approach to develop a drug with desired properties i.e., target protein binding, adverse side effects minimization, and prognosis of BBB penetration. The main advantage of the computer-based approach is target-oriented drug design. The main disadvantage is having limited access to a supercomputer.

## Figures and Tables

**Figure 1 ijms-22-13618-f001:**
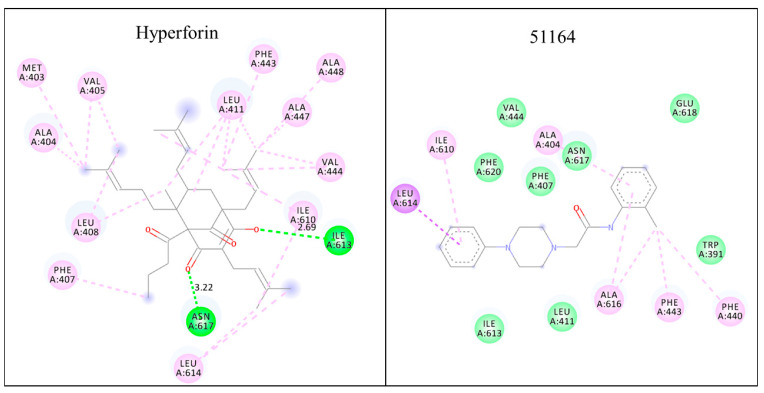
Conformation map of complexation of hyperforin and 51164 with TRPC6. For hyperforin, two hydrogen bonds are visualized with ASN617 (A: ASN617:N2—:Hyp:O) and ILE613 (Hyp:O—A:ILE613:N) with distances of 3.22 Å and 2.69 Å with 64.4° and 45.8° angles, respectively (dark green). All other residues exhibit a hydrophobic type of interaction (purple, alkyl type of bounding). PHE443 exhibits Pi-alkyl type of bounding. For 51164, we find mostly the hydrophobic type of interaction. LEU614 interacts with the Pi-Sigma type to the side aromatic ring of 51164 (A:LEU614:CA—1C6:51164) with an angle deviation of 12.82° and Theta of 8.355 (dark purple). For ALA616, we observed an alkyl type of interaction (3C6:51164: Cl—A:ALA616). PHE443, PHE440, ALA404, and ILE610 hydrophobic residues interacted with the Pi-Alkyl type by maximal 5.45 Å distance. ILE613; LEU411; PHE620; PHE407; ASN617; GLU618 and TRP391 display the van der Waals interaction type (light green).

**Table 1 ijms-22-13618-t001:** List of TRPC6 3D molecular models available at https://www.rcsb.org/.

PDB ID	Description	Refinement Resolution (E)	Method	Global Stoichiometry	Organism	Ref. *
6CV9	Cytoplasmic domain of mTRPC6	3.80	electron microscopy	Homo 4-mer	Mus musculus	[78]
6UZA	Cryo-EM structure of human TRPC6 in complex with antagonist AM-1473	3.08	electron microscopy	Homo 4-mer	Homo sapiens	[76]
6UZ8	Cryo-EM structure of human TRPC6 in complex with agonist AM-0883	2.84	electron microscopy	Homo 4-mer	Homo sapiens	[76]
5YX9	Cryo-EM structure of human TRPC6 at 3.8A resolution	3.80	electron microscopy	Homo 4-mer	Homo sapiens	[71]

* Ref.—reference

**Table 2 ijms-22-13618-t002:** Type of interaction of hyperforin and 51164 with amino acids forming a Ca^2+^-permeable pore of TRPC6.

	Type of Interaction	
Amino Acid of TRPC6	Hyperforin	51164
ALA404	alkyl type, hydrophobic	alkyl type, hydrophobic
LEU411	alkyl type, hydrophobic	van der Waals, electrostatic
PHE443	Pi-alkyl type, hydrophobic	Pi-alkyl type, hydrophobic
ILE610	alkyl type, hydrophobic	Pi-alkyl type, hydrophobic
ILE613	hydrogen bond, donor acceptor	van der Waals, electrostatic
LEU614	alkyl type, hydrophobic	Pi-Sigma, hydrophobic
ASN617	hydrogen bond, donor acceptor	van der Waals, electrostatic

## Data Availability

The data presented in this study are available in Figure 1 and Table 2 of the present study.

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
