# Peer review of "Computer-Based Drug Design of Positive Modulators of Store-Operated Calcium Channels to Prevent Synaptic Dysfunction in Alzheimer’s Disease"

_ijms, 2021, doi:10.3390/ijms222413618_

Round 1

Reviewer 1 Report

Hunanyan et al. in manuscript ijms-1478730 present perspectives to develop modulators of neuronal store-operated calcium entry (nSOCE) based on the available structural information on STIM2, Orai2 and TRPC6. It is suggested that positive modulation of nSOCE may prevent synaptic dysfunction in Alzheimer's disease.

The authors are suggested to please consider the following comments during the revision of the manuscript:

Orai1 mediated local calcium entry is responsible for the activation of NFAT1 which is achieved by association of A-Kinase Anchoring Protein AKAP79 with the N-terminus of Orai1 (Kar et al., 2021, PMID: 33941685). Also, STIM2 has been shown important for NFAT1 activation (Son et al, 2020, PMID: 32601188). In this context it may be worth discussing if there are any perturbations in Calcineurin-NFAT signaling during synaptic dysfunction. Recently it is shown that in addition to inhibition of Orai1-mediated calcium entry by ROS (Bogeski et al., 2010, PMID: 20354224), STIM2-mediated SOCE is also diminished by ROS (Gibhardt et al., 2020, PMID: 33086068). Is there a role of ROS in the pathogenesis of Alzheimer’s disease and can redox modulation of STIM2 in SOCE impact this? Modulation of STIM2 in oxidative stress conditions add complexity in achieveing pharamcological control over STIM2.  

Lines 112-117: The modulation of STIM2 discussed here is not specific to STIM2. This should be clarified. 

Lines 153-155 (Ref# 46): Please provide more details. For examples was Orai2 over-expressed? And whether the effect of Synta66 was assessed on ICRAC or SOCE? Was the effect of Synta66 assessed in absence of endogenous Orai1 and Orai3?

Lines 158-159 (Ref# 47): Some clarity is needed here. Hasn’t this paper rather shown that Orai2/STIM1 ICRAC is smaller than Orai1/STIM1 instead of what is claimed in your statement?

Also, please note that Orai2 has been shown to function as a negative modulator of SOCE in various other cell lines such chondrocytes (Inayama et al., 2015, PMID: 25769459), Jurkat T cells (Alansary et al., 2015, PMID: 25791427), mouse T cells (Vaeth et al., 2017, PMID: 28294127), primary amenoblasts (Eckstein et al., 2019, PMID: 31015290) and human neuroglioma derived cells (Scremin et al., 2020, PMID: 32722509).

Line 166: “Crystal structure” should be changed to “structural information”.  What about the homology models of Orai2?

Section #7: Possible side effects of positive modulation of TRPC6 channel activity - This section may be moved to the end and the authors may want to discuss other targets STIM2/Orai2 in the same context.

Several of the abbreviations are not introduced/expanded, such as BDNF, LTP, etc. Please make sure to do that throughout the text of the manuscript.

Line 214: 5xFAD mouse needs to be described.

The authors describe quite some details on TRPC6 structural information. The readers would benefit if they show a cartoon. Also, the authors should describe a common helix scheme in all TRP channels before drawing comparisons with other TRP channels such as TRPML1.

Section #10: Principles of computer-based drug design - This section should be moved to a much earlier location in the text. Besides that the it just seems to be a list of references. The authors should define some of the processes mentioned.

Line 332: Please refer to the Materials and Methods section.

Conclusion - Lines 395-397: Which experimental evidence is the authors referring to here? Please be more specific and note that the abstract does not reflect what is claimed here.

Line 404: Do you mean having “limited” access?

Author Response

Manuscript Number: IJMS-1478730-R1 

Author’s response: We thank Reviewer 1 for the valuable and constructive comments. We addressed all comments in revised manuscript and provide detailed response below. In order to help reviewers to see the changes we highlighted modified text with yellow color.

 Reviewer 1

Comments and Suggestions for Authors

Hunanyan et al. in manuscript ijms-1478730 present perspectives to develop modulators of neuronal store-operated calcium entry (nSOCE) based on the available structural information on STIM2, Orai2 and TRPC6. It is suggested that positive modulation of nSOCE may prevent synaptic dysfunction in Alzheimer's disease.

The authors are suggested to please consider the following comments during the revision of the manuscript:

Reviewer 1: Orai1 mediated local calcium entry is responsible for the activation of NFAT1 which is achieved by association of A-Kinase Anchoring Protein AKAP79 with the N-terminus of Orai1 (Kar et al., 2021, PMID: 33941685). Also, STIM2 has been shown important for NFAT1 activation (Son et al, 2020, PMID: 32601188). In this context it may be worth discussing if there are any perturbations in Calcineurin-NFAT signaling during synaptic dysfunction.

Author’s response: We thank Reviewer 1 for raising important and critical comment. Indeed in electrically non excitable cells calcineurin-NFAT is main signaling pathway downstream SOCE activation. In neurons it has been shown that calcineurin-NFAT signaling pathway is important for axonal growth during the developmental stage of the brain (Nguyen 2007, PMID: 18093786). However, in adult brain we believe that physiological nSOCE is needed to activate pCaMKII but not calcineurin. In adult 5xFAD (where nSOCE is downregulated) mouse model activation of Calcineurin/NFAT signaling in astrocytes has been shown to drive hyperexcitability (Sompol 2017, PMID: 28559377). In PS1-M146VKI model, nSOCE is downregulated (Sun et al 2014, PMID: 24698269), but calcineurin activity is enhanced, and inhibition of calcineurin activity was neuroprotective (Kim 2015, PMID: 26455952) as well as nSOCE activation (Sun et al 2014, PMID: 24698269). Moreover, there are clinical data confirming that treatment with calcineurin inhibitors may be therapeutic strategy to prevent AD/dementia (Taglialatela, 2015, PMID: 26401556). In addition, there are no papers today showing SOCE mediated calcineurin activation in neurons. Thus, we personally believe that activation of calcineurin-NFAT signaling pathway in adult AD brain is toxic and does not depend on nSOCE, but we do not have experimental observation to state that. We prefer do not discuss research areas where we do not have experimental proofs. In order to keep review focused we would like to skip the discussion of calcineurin-NFAT activation in terms of nSOCE activation.

Reviewer 1: Recently it is shown that in addition to inhibition of Orai1-mediated calcium entry by ROS (Bogeski et al., 2010, PMID: 20354224), STIM2-mediated SOCE is also diminished by ROS (Gibhardt et al., 2020, PMID: 33086068). Is there a role of ROS in the pathogenesis of Alzheimer’s disease and can redox modulation of STIM2 in SOCE impact this?

Author’s response: We thank Reviewer 1 for the comment. ROS plays role in the pathogenesis of AD and it might be that redox modulation of ER-resident protein’s functions such as STIM2, IP3R and RyanR impact function of the nSOCE. It is a great idea to study. However, there are no direct experimental observations has been published so far. So we can only speculate and extrapolate data that have been obtained in non excitable cells (such as immune T cells, Gibhardt et al., 2020, PMID: 33086068) to neurons. Again role of mitochondria and ROS in the pathogenesis of AD is not the field of our expertise and we feel uncomfortable to discuss it. Moreover, since the target is unknown the discussion of this topic wont bring us novel therapeutic compounds that would be able to normalize nSOCE function in the brain.

Reviewer 1: Modulation of STIM2 in oxidative stress conditions add complexity in achieveing pharamcological control over STIM2.  

Author’s response: We agree the statement of the Reviewer 1. We believe that oxidative stress impacts functions of STIM2 and other partners of nSOCE, especially at late stages of the disease. Whether ROS impacts function of nSOCE in early stages (when application of nSOCE correcting drugs would be most effective) of the AD is the matter of debate and the discussion of this topic will bring us far from the focus of the review.

Reviewer 1: Lines 112-117: The modulation of STIM2 discussed here is not specific to STIM2. This should be clarified. 

Author’s response: we have incorporated word “unspecific” in line 133

Reviewer 1: Lines 153-155 (Ref# 46): Please provide more details. For examples was Orai2 over-expressed? And whether the effect of Synta66 was assessed on ICRAC or SOCE? Was the effect of Synta66 assessed in absence of endogenous Orai1 and Orai3?

Author’s response: We thank Reviewer 1 for constructive criticism and provided requested info as following:

Lines 176-178

So far only one chemical compound, Synta66 (N-[4-(2,5-dimethoxyphenyl)phenyl]-3-fluoropyridine-4-carboxamide), has been shown to potentiate ORAI2- dependent calcium release activated calcium (CRAC) channel current density in HEK293T cells overexpressing CFP-tagged ORAI2 on the ORAI1/2/3 null background

Reviewer 1: Lines 158-159 (Ref# 47): Some clarity is needed here. Hasn’t this paper rather shown that Orai2/STIM1 ICRAC is smaller than Orai1/STIM1 instead of what is claimed in your statement?

Author’s response: We agree with Reviewer 1 that paper with Ref#54 (former Ref#47)shows that Orai2/STIM1 ICRAC is smaller than Orai1/STIM1. However, the paper also shows that overexpression of ORAI2 without STIM1 (Fig 3B, Ref 54, open square versus filled triangle) significantly reduced SOCE amplitude HEK cells. Therefore, we suggest that our statement is based on published data and reliable to cite as it is.

Reviewer 1: Also, please note that Orai2 has been shown to function as a negative modulator of SOCE in various other cell lines such chondrocytes (Inayama et al., 2015, PMID: 25769459), Jurkat T cells (Alansary et al., 2015, PMID: 25791427), mouse T cells (Vaeth et al., 2017, PMID: 28294127), primary amenoblasts (Eckstein et al., 2019, PMID: 31015290) and human neuroglioma derived cells (Scremin et al., 2020, PMID: 32722509).

Author’s response: We thank Reviewer 1 for the comment. Orai2 mediated negative regulation of SOCE in Jurkat T cells is discussed in the paper (Ref #57, Vaeth et al., 2017, PMID: 28294127)). The rest mentioned references are added to the text on lines 184-185.

Reviewer 1:  Line 166: “Crystal structure” should be changed to “structural information”.  What about the homology models of Orai2?

Author’s response: “Crystal structure” is changed to “structural information” on line 192. Currently, the Swiss-Prot and TrEMBL databases contain 40 molecular models of ORAI2 obtained on the basis of homologous modeling for different organisms; however, there is no complete three-dimensional homologous protein model. One of the successful models presented in Swiss-Prot with identification number Q96SN7 (ORAI2_HUMAN) has an Average Model Confidence (QMEANDisCo) value of 0.59 ± 0.05 with identification of amino acid sequences at 58.38%. Another model available in TrEMBL (C9JQR7_HUMAN) has an Average Model Confidence (QMEANDisCo) = 0.58 ± 0.05, while the amino acid sequence identification value is 61.93%. Sequence identification value around 60% would rather give bad quality model that would not be appropriate for drug design. Besides, we argue that ORAI2 is not well defined functionally both in neuronal and non neuronal cells (lines 171-173, 179-182) thus even if we get a chemical compound we would not be able to test it experimentally.

We have added short information on available but not appropriate for drug design homology models of ORAI2 on line 192-193.

Reviewer 1:  Section #7: Possible side effects of positive modulation of TRPC6 channel activity - This section may be moved to the end and the authors may want to discuss other targets STIM2/Orai2 in the same context.

Author’s response: We would like to leave the section #8 (former section #7) in place where it is. We would not like to finish our paper by the section where possible side effects are discussed since we believe it would bring negative impression from the paper and would rase the questions among readers on reasonabilities to continue studies. We would not like to describe possible side effects of positive modulators of STIM2/ORAI2 proteins since possible side effects should be clear to readers from the sections #4 and #6.

Reviewer 1:  Several of the abbreviations are not introduced/expanded, such as BDNF, LTP, etc. Please make sure to do that throughout the text of the manuscript.

Author’s response: We thank Reviewer 1 for careful reading and apologize for lack of decryptions of some abbreviations. We introduced LTP, LTD, CRAC, BDNF and etc

Reviewer 1:  Line 214: 5xFAD mouse needs to be described.

Author’s response: Definition of 5xFAD mice is introduced in lines 241-242.

Reviewer 1: The authors describe quite some details on TRPC6 structural information. The readers would benefit if they show a cartoon. Also, the authors should describe a common helix scheme in all TRP channels before drawing comparisons with other TRP channels such as TRPML1.

Author’s response: We thank Reviewer 1 for valuable comment. However, we believe that addition of a cartoon describing common helix scheme in all TRP channels would further complicate and increase volume of the paper and would probably mislead the readers. TRP protein family includes many members, many structural and bioinformatical data are obtained in a way that they can not be directly compared i.e. structure is provided for monomer or for a tetramer.

In order to help readers to understand main message of the paper we decided to remove the information about TRPML1 and CavAb from the text.

Reviewer 1: Section #10: Principles of computer-based drug design - This section should be moved to a much earlier location in the text. Besides that the it just seems to be a list of references. The authors should define some of the processes mentioned.

Author’s response: We thank Reviewer 1 for the comment. We have changed section #10 to section #2 and added following descriptions:

Lines 86-96

These processes are: virtual screening which is a compound selection process that calcu-lates structure similarity values based on quantitative structure-property (QSPR) and structure-activity (QSAR) characteristics, using different similarity descriptors [23], phar-macophore design which is a procedure for determining the sets of compound’s spatial-energy characteristics necessary to ensure high affinity complexation of the compound with biological target, leading to a change in target’s biological response [18, 24], dynamic modeling is one of the main techniques that allows one to simulate the interactions of the target-ligand system in real time [25], chemogenomics and proteomics, using these meth-ods for the design of pharmaceutical compounds, researchers are studying genomic and / or proteomic responses to compounds of various natures in biomedical molecular sys-tems [26].

Reviewer 1:  Line 332: Please refer to the Materials and Methods section.

Author’s response: Line 344 - Materials and Methods are referred

Reviewer 1:  Conclusion - Lines 395-397: Which experimental evidence is the authors referring to here? Please be more specific and note that the abstract does not reflect what is claimed here.

Author’s response: We have added experimental details that support TRPC6 as a perspective pharmacological target as following: “decreased TRPC6 mRNA expression in AD patient samples, improved cognitive function and increased excitatory synapse formation in mice that overexpress TRPC6 as well as synaptoprotective properties of positive TRPC6 modulators” (Lines 421-423).

In addition we provided arguments on why  STIM2 and ORAI2 are not appropriate target to correct AD pathology (lines 425-427). We hope that in the current form conclusion is better correlated with the abstract.

Reviewer 1:  Line 404: Do you mean having “limited” access?

Author’s response: We are sorry for confusing sentence. Yes, we meant limited access. The sentence is corrected (Line 436).

Reviewer 2 Report

The manuscript of Hunanyan et al. aim to illustrate the possibility of developing modulators of neuronal store-operated channels on available structures of TRP6, ORAI2 and STIM2. They discuss the reasons why STIM2 and ORAI2 could not be considered good lead cellular targets for drug design and concentrate on the characteristics of TRP6 modulators. They apply the docking and conformation analysis to both hyperforin and 51164 compound. The first part of the article sections 1-11 is well written, but to my opinion, the last part which includes the own work of the authors is not well integrated into the paper. I suggest to include the section 12 in section 11, maybe as a subsection. Although the authors claim that “hey  provide commonly used steps applied for drug design based on 3D structures of target proteins that might be used to  develop novel AD preventing therapy”,  I consider that the part concerning  how the information obtained from conformation analysis could be the basis to further drug design studies is very poorly described (356-368), and should be rewritten in order not only to harmonise the style with the rest of the paper, but also to illustrate the perspectives of the work.  Similarly, they conclude that computer based drug design might be a tool to develop  drugs “with desired properties i.e., target protein binding, adverse side effects minimization  and prognosis of brain blood barrier  penetration”, without having provided information about the latter points. The last sentences in the conclusions “The main disadvantage is having access to a supercomputer” should be rewritten.

The first part of the article sections 1-11 is well written, but to my opinion, the last part which includes the own work of the authors is not well integrated into the paper. I suggest to include the section 12 in section 11, maybe as a subsection, and to rewrite the last part (355-368), not only in order to armonize the style with the rest of the paper, but also  to illustrate the perspectives of the work, considering that the article is a review. For example it is surprising to read “Determination of top 15 compounds based on in silico screening results” without any argumentation.

Minor points:

Some sentences should be rewritten and some typos should be corrected. I enclose some examples:

Sentences difficult to understand:

55 “we will focus on the development of positive modulators of nSOCE intracellular signaling pathway due to the fact there is no preclinical  model available so for that would allow to test cognitive benefit of usage of nSOCE blockers”

69 “Major advantage of in silico drug design that it starts from the molecular target, those  function is disrupted in the disease, and searches chemical compounds that are able to  normalize either structure or function of the target protein”

Please write consistently in the whole paper hyperforin or Hyperforin

123 Isoforms of STIM2 are  differ in the length of signal peptide at the N terminus

216 Interestingly that

218 It was reported that this compound was effective at low-micromolar concentrations and demonstrate

241 This molecular model is released in 2018.

242 It should be noted that  TRPC6 is potential drug targets included in the

Abbreviations should be defined the first time they are used: 141 PDB; 276 n CavAb)

348 interraction

365 Determination of top 15 compounds based on in silico screening results. It is not clear for the reader  why 15 instead of 5 or 50. The Authors should consider that the article is a review.  

329 we describe or own study of 51164 compound

402 with desire properties

References: number 5 and  number 24 are identical;  incorrect year in ref. 85

Author Response

Manuscript Number: IJMS-1478730-R1 

Author’s response: We thank Reviewer 2 for careful reading of the manuscript and the constructive evaluation of our submission. We addressed these comments in a revised manuscript and provide point-by-point answers below. To help Reviewer see the changes we highlighted text with yellow color

Reviewer 2

Comments and Suggestions for Authors

The manuscript of Hunanyan et al. aim to illustrate the possibility of developing modulators of neuronal store-operated channels on available structures of TRP6, ORAI2 and STIM2. They discuss the reasons why STIM2 and ORAI2 could not be considered good lead cellular targets for drug design and concentrate on the characteristics of TRP6 modulators. They apply the docking and conformation analysis to both hyperforin and 51164 compound.

Reviewer 2:  The first part of the article sections 1-11 is well written, but to my opinion, the last part which includes the own work of the authors is not well integrated into the paper. I suggest to include the section 12 in section 11, maybe as a subsection. Although the authors claim that “hey  provide commonly used steps applied for drug design based on 3D structures of target proteins that might be used to  develop novel AD preventing therapy”,  I consider that the part concerning  how the information obtained from conformation analysis could be the basis to further drug design studies is very poorly described (356-368), and should be rewritten in order not only to harmonise the style with the rest of the paper, but also to illustrate the perspectives of the work.  Similarly, they conclude that computer based drug design might be a tool to develop  drugs “with desired properties i.e., target protein binding, adverse side effects minimization  and prognosis of brain blood barrier  penetration”, without having provided information about the latter points.

Author’s response: We thank Reviewer 2 for valuable comment. Please find rewritten text on lines 377-397.

Reviewer 2:  The last sentences in the conclusions “The main disadvantage is having access to a supercomputer” should be rewritten.

Author’s response: We are sorry for misleading sentence. We have added word ‘limited’ to the sentence (line 436).

Reviewer 2:  The first part of the article sections 1-11 is well written, but to my opinion, the last part which includes the own work of the authors is not well integrated into the paper. I suggest to include the section 12 in section 11, maybe as a subsection, and to rewrite the last part (355-368), not only in order to armonize the style with the rest of the paper, but also  to illustrate the perspectives of the work, considering that the article is a review. For example it is surprising to read “Determination of top 15 compounds based on in silico screening results” without any argumentation.

Author’s response: We thank Reviewer 2 for a constructive criticism. We have incorporated section 12 in section 11 as a subsections (11.1, 11.2, 11.3) and have rewritten last part completely in order to harmonize last section with the rest of the paper.

Please see changes highlighted by yellow color on lines 377-397.

Reviewer 2:  Minor points:

Some sentences should be rewritten and some typos should be corrected. I enclose some examples:

Author’s response: We are sorry for complicated sentences and typos. We have tried to simplify our writing as well as corrected typos.

Reviewer 2:  Sentences difficult to understand:

55 “we will focus on the development of positive modulators of nSOCE intracellular signaling pathway due to the fact there is no preclinical  model available so for that would allow to test cognitive benefit of usage of nSOCE blockers”

Author’s response: We are sorry for confusing writing. We have changed the description as following:

Lines 55-58: However, there is no preclinical model available so for that would allow to test cognitive benefit of usage of nSOCE antagonists. Thus, within the current paper we will focus on the development of positive modulators of nSOCE intracellular signaling pathway.

Reviewer 2:  69 “Major advantage of in silico drug design that it starts from the molecular target, those  function is disrupted in the disease, and searches chemical compounds that are able to  normalize either structure or function of the target protein”

Author’s response: We are sorry for confusing writing. We have changed the description as following:

Line 69:

Major advantage of in silico drug design - search and design of chemicals that will normal-ize either structure or function of the target protein.

Reviewer 2: Please write consistently in the whole paper hyperforin or Hyperforin

Author’s response: We are sorry for inconsistent writing, we have changed to Hyperforin in the whole paper.

Reviewer 2: 123 Isoforms of STIM2 are differ in the length of signal peptide at the N terminus

Author’s response: Line 145 - Dash is inserted in N-terminus

Reviewer 2: 216 Interestingly that

Author’s response: Line 245 - “that” is deleted

Reviewer 2: 218 It was reported that this compound was effective at low-micromolar concentrations and demonstrate

Author’s response: Line 248 – typo is corrected

Reviewer 2: 241 This molecular model is released in 2018.

Author’s response: Line 270 – “This” is substitutes by “6CV9”

Reviewer 2: 242 It should be noted that  TRPC6 is potential drug targets included in the

Author’s response: Line 272 – typo (targets) is corrected

Reviewer 2: Abbreviations should be defined the first time they are used: 141 PDB; 276 n CavAb)

Author’s response: Abbreviations are defined: Line 163 – protein data bank (PDB), CavAb is deleted from the revised version of the text.

Reviewer 2: 348 interraction

Author’s response: Line 364-Mistake is corrected (interraction)

Reviewer 2: 365 Determination of top 15 compounds based on in silico screening results. It is not clear for the reader  why 15 instead of 5 or 50. The Authors should consider that the article is a review.

Author’s response: We are sorry for the confusion. We have deleted quantity of compounds. Instead, we provide the characteristics that apply to the top connections.

Reviewer 2: 329 we describe or own study of 51164 compound

Author’s response: Line 345 – typo is corrected (our own)

Reviewer 2: 402 with desire properties

Author’s response: The sentence has been deleted during revision process.

Reviewer 2: References: number 5 and  number 24 are identical;  incorrect year in ref. 85

Author’s response: We are sorry for double citation and wrong year in ref 85. Reference 24 is substituted by reference 5 (currently it is ref #6). Year 1998 is inserted into reference 85.

Round 2

Reviewer 1 Report

My concerns have been addressed and I do not have any new comments.